# Longitudinal gut microbiota composition of South African and Nigerian infants in relation to tetanus vaccine responses

Saori C. Iwase,[1,2] Sophia Osawe,[3] Anna-Ursula Happel,[1,2] Clive M. Gray,[4] Susan P. Holmes,[5] Jonathan M. Blackburn,[2,6] Alash'le Abimiku,[3,7] Heather B. Jaspan[1,2,8]

**ABSTRACT**  Infants who are exposed to HIV but uninfected (iHEU) have higher risk of infectious morbidity than infants who are HIV-unexposed and uninfected (iHUU), possibly due to altered immunity. As infant gut microbiota may influence immune development, we evaluated the effects of HIV exposure on infant gut microbiota and its association with tetanus toxoid vaccine responses. We evaluated the gut microbiota of 82 South African (61 iHEU and 21 iHUU) and 196 Nigerian (141 iHEU and 55 iHUU) infants at <1 and 15 weeks of life by 16S rRNA gene sequencing. Anti-tetanus antibodies were measured by enzyme-linked immunosorbent assay at matched time points. Gut microbiota in the 278 included infants and its succession were more strongly influenced by geographical location and age than by HIV exposure. Microbiota of Nigerian infants, who were exclusively breastfed, drastically changed over 15 weeks, becoming dominated by *Bifidobacterium longum* subspecies *infantis*. This change was not observed among South African infants, even when limiting the analysis to exclusively breastfed infants. The Least Absolute Shrinkage and Selection Operator regression suggested that HIV exposure and gut microbiota were independently associated with tetanus titers at week 15, and that high passively transferred antibody levels, as seen in the Nigerian cohort, may mitigate these effects. In conclusion, in two African cohorts, HIV exposure minimally altered the infant gut microbiota compared to age and setting, but both specific gut microbes and HIV exposure independently predicted humoral tetanus vaccine responses.

**IMPORTANCE**  Gut microbiota plays an essential role in immune system development. Since infants HIV-exposed and uninfected (iHEU) are more vulnerable to infectious diseases than unexposed infants, we explored the impact of HIV exposure on gut microbiota and its association with vaccine responses. This study was conducted in two African countries with rapidly increasing numbers of iHEU. Infant HIV exposure did not substantially affect gut microbial succession, but geographic location had a strong effect. However, both the relative abundance of specific gut microbes and HIV exposure were independently associated with tetanus titers, which were also influenced by baseline tetanus titers (maternal transfer). Our findings provide insight into the effect of HIV exposure, passive maternal antibody, and gut microbiota on infant humoral vaccine responses.

**KEYWORDS**  HIV-exposed uninfected infants, South Africa, Nigeria, gut microbiota, tetanus toxoid, vaccine response

The mutualistic relationship between microbes and humans begins in early life. Emerging evidence suggests that the colonization of microbes in the gut facilitates the development of the immune system and growth trajectories (1, 2). Due to the successful prevention of vertical transmission programs, the number of infants who are

Address correspondence to Heather B. Jaspan, HBJaspan@gmail.com.

Saori C. Iwase and Sophia Osawe contributed equally to this article. Order of authors was determined on primary manuscript contribution.

The authors declare no conflict of interest.

See the funding table on p. 14.

HIV-exposed yet uninfected (iHEU) has been increasing, particularly in sub-Saharan Africa (3). Compared to infants who are HIV-unexposed and uninfected (iHUU), iHEU are at higher risk of morbidity and mortality, predominantly due to infectious diseases (4). This is thought to be linked to their altered immunity (5, 6), which may be secondary to altered gut microbiota. To our knowledge, there are limited longitudinal studies comparing gut microbiota between iHEU and iHUU, and most of them were conducted in a single country (7–10). Some studies have found few differences (8, 10, 11), whereas clear differences in microbiota profile were observed in Haitian (7) and Nigerian iHEU (9). Thus far, only one cross-sectional study compared the gut microbiota between iHEU and iHUU in multiple countries, including Belgium, Canada, and South Africa, and suggested that the difference in microbiota by HIV exposure status may be population-specific (12). Therefore, the effect of geography and HIV exposure on infant gut microbiota requires further investigation.

Vaccines are critical for protecting infants from infectious diseases and consequent morbidity and mortality. However, multiple factors can influence vaccine immunogenicity, including genetics, nutritional status, and pre-existing immunity (13). In addition, emerging evidence points to a possible role of the gut microbiome in influencing vaccine response (14). In Bangladeshi infants, CD4+ T cell proliferation and IgG against tetanus toxoid (TT) vaccination were positively associated with abundance of Actinobacteria, particularly *Bifidobacterium longum,* until at least 2 years of age (15, 16). Conversely, vaccine-induced CD4+ T cell proliferation against TT vaccine was negatively associated with the abundance of *Enterobacteriales* and *Pseudomonadales* (15).

To evaluate the contribution of gut microbiota to observed differences in immunity between iHEU and iHUU, we longitudinally compared the gut microbiota of South African and Nigerian infants exposed and unexposed to HIV, and correlated these with TT vaccine responses.

## MATERIALS AND METHODS

### Study participants

Mothers with and without HIV and their neonates were recruited into a multicenter longitudinal study between September 2013 and November 2017 (17). Mother-infant pairs were enrolled during the first week post-delivery at the Khayelitsha Site B Midwife Obstetric Unit in Cape Town, South Africa, and the Plateau State Specialist Hospital in Jos, Nigeria. Clinical and demographic data and samples (including stool and blood) were collected. All mothers with HIV received antiretroviral therapy according to local guidelines, and their infants were confirmed as HIV negative by polymerase chain reaction (PCR) at birth and later time points (18, 19). In addition, all iHEU received nevirapine post-exposure prophylaxis after birth, and co-trimoxazole was recommended at 6 weeks of age as per country-specific guidelines (18, 20). Exclusive breastfeeding was advised to all mothers for 6 months. Feeding data were collected using a structured questionnaire validated in similar settings (21). Feeding practices were categorized as "exclusive breastfeeding," defined as receiving only breastmilk or prescribed medicines since birth, or "mixed feeding," defined as receiving breastmilk supplemented with other liquids or food or receiving formula. In this analysis, we included stool and plasma collected from term infants during the first and at 15 weeks of life born to mothers without complications during pregnancy or delivery.

### Immunization

Routine childhood vaccinations were given to all infants according to the World Health Organization Expanded Program on Immunization (22). In both countries, infants were vaccinated against tetanus at 6, 10, and 14 weeks. South African infants received DTaP [diphtheria toxoid (DT), TT, and acellular pertussis (aP)], while Nigerian infants received DTwP [DT, TT, and whole-cell pertussis (wP)]. Pregnant mothers were given booster TT vaccination (Serum Institute of India Pvt. Ltd.) in Nigeria.

## Sample collection, DNA extraction, and 16S rRNA gene sequencing

Fecal samples were collected from diapers, avoiding the surface. Samples were placed on ice immediately, transferred to the lab within 6 hours, and stored at −40 to −20°C until analysis. Samples were thawed and treated with a cocktail of mutanolysin (300 U/mL, Sigma Aldrich), lysozyme (45,000 U/mL, Sigma Aldrich), and lysostaphin (24 U/mL, Sigma Aldrich) in 300 µL phosphate buffered solution (PBS) for 1 hour at 37°C. Samples were then mechanically disrupted by bead-beating at 50 Hz for 10 minutes using the Qiagen TissueLyser LT (23). DNA was extracted using the PowerSoil DNA extraction kit (Qiagen), following the manufacturer's protocol. For cross-contamination filtering, genomic DNA was extracted from mock bacterial community cells with equal colony-forming units from each of the 22 known species (HM-280, Biodefense and Emerging Infections Research Resources Repository [BEI]). Extracted genomic DNA was subjected to PCR amplification in triplicate using primers targeting the V3–V4 region (357F/806R primers) of the 16S rRNA gene, as described previously (24). Negative controls for DNA extraction and PCR were included. Amplified libraries were purified using AMPure XP beads (Beckman Coulter), quantitated using Quant-iT dsDNA High Sensitivity Assay Kits (ThermoFisher), pooled in equal molar amounts, and paired-end sequenced using a MiSeq Reagent Kit V3 (600-cycle, Illumina). Following demultiplexing, barcode primers were removed using Cutadapt (version 3.4) (25), and reads were processed using DADA2 (version 1.19.2) (26) within the R framework (R version 4.0.4) (27). Taxonomic classification of amplicon sequence variants (ASVs) was done using an updated SILVA training set (version 132) (28), available at https://github.com/itsmisterbrown/updated_16S_dbs (29). Contaminant ASVs were identified and removed using the decontam package (version 1.16.0) (30). Samples with less than 2,000 filtered reads were excluded from the downstream analysis.

## Plasma IgG anti-tetanus antibodies

Blood samples were obtained from infants at 1 and 15 weeks and from Nigerian mothers at 1 week postpartum. All blood samples were collected in heparinized tubes and transported to the lab within 6 hours for sample processing. Plasma was removed prior to cell isolation using Ficoll density-gradient separation medium (Sigma Aldrich) and stored at −80°C until analysis. Plasma anti-tetanus IgG was measured by enzyme-linked immunosorbent assay (ELISA) following the manufacturer's protocol (TECAN, IBL International GmbH). The optical density at 450 nm was measured using an ELISA microplate reader (BioTek ELx808 absorbance plate reader), and a standard curve was generated using the readings from the calibrators included on each plate and used to calculate the individual titers (IU/mL). To validate each assay, we considered only calibration curves for each plate with a coefficient of determination ($r^2$) above 0.95. Calibration curves that had an $r^2$ below 0.95 were repeated. The manufacturer provided the intra-assay and inter-assay coefficient of variation (CV%) as 6.9 and 10.4, respectively. Samples on each plate were run in duplicate and averaged. Previously tested positive samples were incorporated into subsequent runs as in-house controls.

## Data analysis

Differences in study cohort characteristics were assessed using the Student's $t$-test (parametric continuous variables), Wilcoxon signed-rank test (non-parametric continuous variables), and $\chi^2$ test (parametric categorical variables). Spearman's rank correlation coefficient ($R$) was used to analyze associations between groups. Bacterial community analysis was done using the phyloseq (version 1.40.0) (31) and vegan (version 2.4.6) (32) packages. The ASV table was normalized (i.e., transformed to relative abundance * median sample read depth) and filtered so that each ASV had at least 10 counts in at least 20% of the samples or had a total relative abundance of at least 0.1%. Shannon index was calculated as a measure of α-diversity. Comparison of microbial community composition between groups was evaluated by principal coordinate analysis (PCoA) and

permutational multivariate analysis of variance (PERMANOVA) using the adonis2 function in the vegan package (32), based on the Bray-Curtis dissimilarity and 999 permutations. Partitioning around medoids (PAM) clustering was applied to determine the optimal $k$ using the cluster package (version 2.1.4) (33). Analysis of Compositions of Microbiomes with Bias Correction (ANCOM-BC; version 1.6.4) (34) was used to identify significantly differentially abundant ASVs through pair-wise comparisons with an adjusted $P$-value of <0.05, and $\log_e$ fold change of >0.5 or <−0.5. $P$-values of anti-tetanus IgG titers were compared by Wilcoxon signed-rank tests and adjusted for multiple comparisons using the Benjamini-Hochberg method. To identify factors associated with infant anti-tetanus IgG titers at 15 weeks of age, we applied the Least Absolute Shrinkage and Selection Operator (LASSO) regression using the glmnet package (version 4.1.4) (35). Since microbiome compositional data are often highly skewed, we employed rank-based transformation (36) for the regression analysis using the top 50 ASVs among infants who had microbiota data available at both week 1 and week 15. After the transformation, the most abundant bacterial taxon within the sample was given the highest score of 50, and the least abundant bacterial taxon was given a score of 1. The rank-transformed ASVs, infant anti-tetanus IgG titers at 1 week of age (indicative of passive maternal antibody transfer), and HIV exposure status were used as explanatory variables. Models were created according to the infant's age and geographical location separately. The predictive models were validated by 10-fold cross-validation using the cv.glmnet() function in the glmnet package (35). Lambda value that gave the lowest model error was used as a tuning parameter. Variables that fitted within the regression model were considered to be predictor variables for the TT vaccine response. $P$-values <0.05 and 95% confidence intervals were used to assess statistical significance.

## RESULTS

### Cohort characteristics

Overall, there were 278 mother-infant pairs included in this analysis; 82 were from South Africa and 196 were Nigerian. Several demographic and socioeconomic characteristics differed by study site (Table 1). At enrolment, Nigerian mothers were older [mean age 31 (standard deviation (SD) ±5.31) versus 28 (SD ±5.38) years, $P = 0.001$] with higher gravidity [median 2 (interquartile range (IQR): 1–4) versus 1 (IQR: 1–2), $P < 0.001$] and lower body weight [mean 62.87 (SD ±11.51) versus 72.69 (SD ±13.86) kg, $P < 0.001$] than South African mothers. While electricity was equally available for participants from both countries, significantly more mothers in South Africa had a refrigerator and running water at home, and significantly more Nigerian mothers lived in formal housing (all $P$ < 0.001). The weight-for-length z score (wflz) of Nigerian infants was significantly lower than that of South African infants at 15 weeks of age (0.54 versus 0.86, $P = 0.023$). All Nigerian infants were exclusively breastfed (EBF) until 15 weeks of life, whereas only 58.5% of South African mothers reported still EBF at 15 weeks postpartum ($P < 0.001$). Among the South African mothers who reported "mixed feeding," 58.8% ($n = 19$) introduced formula feeding or solid food while continuing breastfeeding, and 41.2% ($n = 14$) completely switched to formula feeding during the course of the study (median breastfeeding duration: 32 days). Mothers of iHUU had higher formal education than mothers of iHEU ($P = 0.002$; Table S1). History of antibiotic use was higher among iHEU due to co-trimoxazole prophylaxis (86.6% iHEU versus 6.6% iHUU, $P < 0.001$). Significantly fewer South African iHEU reported co-trimoxazole prophylaxis than Nigerian iHEU (55.7% versus 100%, $P < 0.001$).

### Gut microbiota differs substantially between South African and Nigerian infants in the first week of life

Of the 524 samples sequenced, 442 samples passed the quality filtering of requiring at least 2,000 filtered reads. Thus, 164 (47 South African and 117 Nigerian) out of the 278 infants had gut microbiota data available at both 1 and 15 weeks of life. Gut microbiota

**TABLE 1** Cohort characteristics[c]

| | | South Africa (N = 82) | Nigeria (N = 196) | P |
|---|---|---|---|---|
| Maternal characteristics | | | | |
| Mother's age at delivery [years; mean (SD)] | | 28 (5.38) | 31 (5.31) | 0.001 |
| Education (n; %) | None | 0 (0.0) | 2 (1.0) | <0.001 |
| | Elementary | 5 (6.1) | 65 (33.2) | |
| | Secondary | 72 (87.8) | 73 (37.2) | |
| | Higher | 5 (6.1) | 56 (28.6) | |
| Unemployed (n; %) | | 59 (72.0) | 6 (3.1) | <0.001 |
| Formal housing (n; %) | | 33 (40.2) | 185 (94.4) | <0.001 |
| Electricity (n; %) | | 78 (95.1) | 178 (90.8) | 0.332 |
| Refrigerator (n; %) | | 70 (85.4) | 96 (49.0) | <0.001 |
| Running water (n; %) | | 38 (46.3) | 48 (24.5) | 0.001 |
| Marital status (n; %) | Married/living together | 25 (30.5) | 186 (94.9) | <0.001 |
| | Single | 57 (69.5) | 10 (5.1) | |
| Gravidity [n; median (IQR)] | | 1 [1, 2] | 2 [1, 4] | <0.001 |
| Mother's weight at enrollment [kg; mean (SD)][a] | | 72.69 (13.86) | 62.87 (11.51) | <0.001 |
| Infant characteristics | | | | |
| iHEU (n; %) | | 61 (74.4) | 141 (71.9) | 0.787 |
| Male (n; %) | | 41 (50.0) | 94 (48.0) | 0.858 |
| Gestational age at delivery [weeks; median (IQR)] | | 39.30 [38.02, 40.38] | 39.95 [38.98, 40.62] | 0.011 |
| Vaginal delivery (n; %) | | 82 (100.0) | 167 (85.2) | 0.001 |
| Wflz at W15 [median (IQR)][b] | | 0.86 [0.32, 1.90] | 0.54 [−0.64, 1.42] | 0.023 |
| Mode of feeding at W15 (n; %) | Exclusive breastfeeding | 48 (58.5) | 196 (100.0) | <0.001 |
| | Mixed feeding | 34 (41.5) | 0 (0.0) | |
| Reported antibiotic use (n; %) | Co-trimoxazole | 34 (41.5) | 141 (71.9) | <0.001 |
| | Other | 2 (2.4) | 3 (1.5) | |

[a]Missing data from five Nigerian participants.
[b]Missing data from 41 participants (South Africa, n = 16; Nigeria, n = 25).
[c]IQR, interquartile range; SD, standard deviation; iHEU, infants who are HIV-exposed yet uninfected; W15, 15 weeks of age; wflz, weight-for-length z score.

composition differed significantly by study site during the first week of life (Fig. 1A). Within-sample microbial diversity (Shannon index) was higher among South African than Nigerian infants [median 2.23 (IQR: 1.96–2.62) versus 1.13 (IQR: 0.77–1.68), $P < 0.0001$; Fig. 1B]. In addition, microbial community composition was significantly different by geographical location, although the site only explained 6% of the community composition (Fig. 1C; PERMANOVA $P < 0.001$). Geographical location remained significantly associated with α- and β-diversity after adjusting for sequencing batch or in separate models adjusting for demographic factors that significantly differed between countries, namely maternal marital status, weight, age, gravity, education level, occupation, type of house, access to a refrigerator or running water, mode of delivery, or infant gestational age ($P < 0.001$ for both α- and β-diversity). Moreover, α- and β-diversity remained significantly different by the geographic location when the comparison was made strictly among samples collected on the first day of life ($n = 147$; Fig. S1).

At baseline, most South African infants had gut microbiota consisting of (i) Actinomycetota, including several *Bifidobacterium* species (such as *B. longum* subspecies *longum*, *Bifidobacterium catenulatum*, and *Bifidobacterium breve*) and *Collinsella aerofaciens*, (ii)

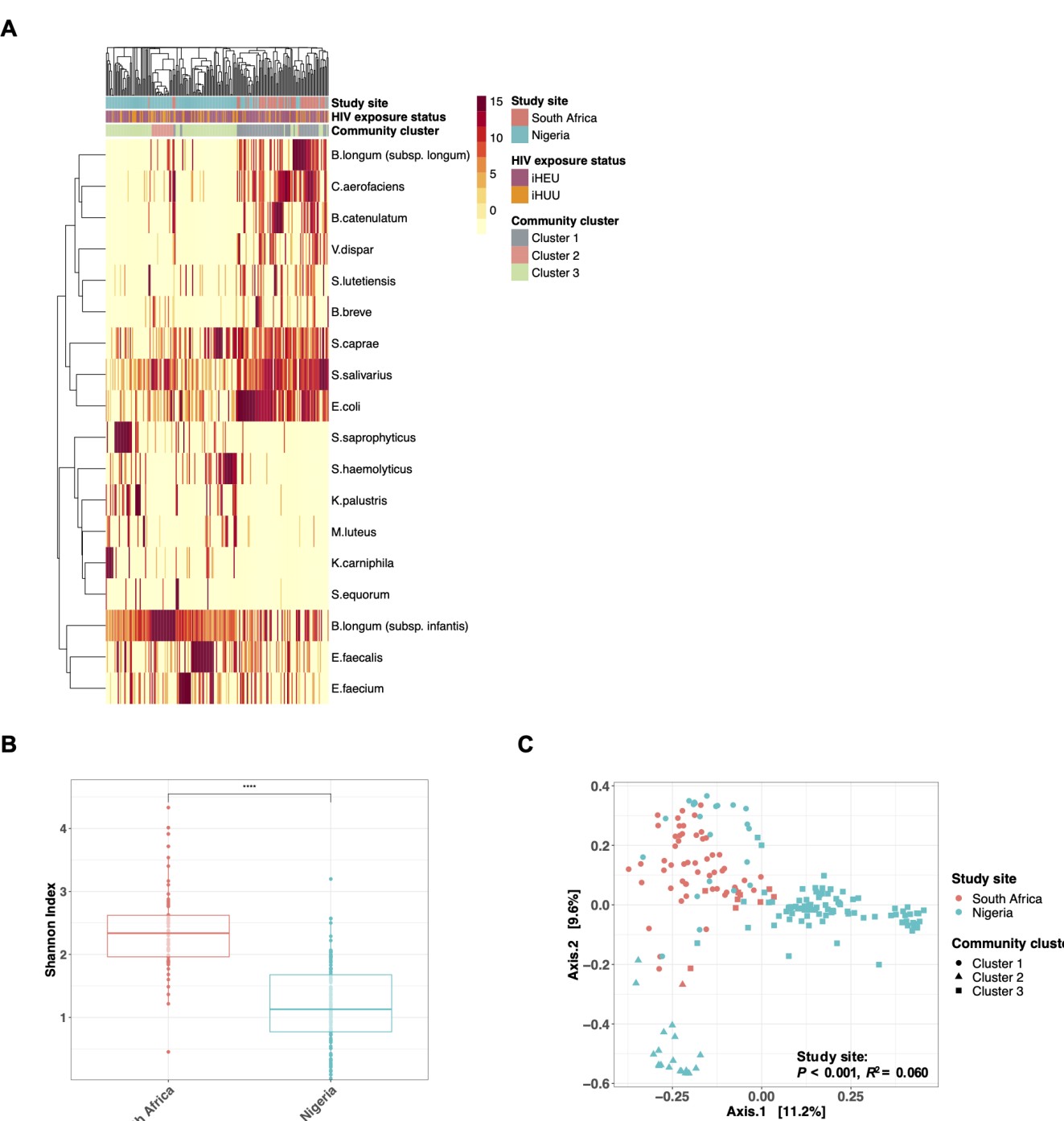

**FIG 1** Geographical location strongly affects gut microbiota among African infants in the first week of life. (A) Heatmap of the top 20 taxa in the gut microbiota of South African ($n = 63$) and Nigerian ($n = 141$) infants in the first week of age. Study site, HIV exposure status, and community cluster types (based on PAM clustering; $k = 3$) are shown in annotation bars. (B) Comparison of α-diversity (Shannon index) between South African ($n = 63$) and Nigerian ($n = 141$) infants during the first week of life. (C) PCoA and PERMANOVA (Bray-Curtis dissimilarity) of gut microbiota during the first week of age (South African, $n = 63$; Nigerian, $n = 141$), colored by study site and shaped by community groups, based on PAM clustering ($k = 3$). ****$P < 0.0001$.

Firmicutes, including *Streptococcus* species (such as *Streptococcus salivarius*, *Streptococcus caprae*, and *Streptococcus lutetiensis*) and *Veillonella dispar,* and (iii) Proteobacteria which mainly consist of *Escherichia coli*, which was named "cluster 1" identified by PAM

clustering (Fig. 1A). On the other hand, the majority of Nigerian infants' microbiota was classified as community cluster 3, dominated by (i) Actinomycetota, mainly *B. longum* subspecies *infantis* and (ii) Firmicutes, including *Staphylococcus* species (such as *Staphylococcus haemolyticus* and *Staphylococcus saprophyticus*) and *Enterococcus* species (such as *Enterococcus faecalis* and *Enterococcus faecium*).

## Age is a major driver of microbiota development, but microbial succession differs between sites

We next assessed gut microbiota longitudinally. The α-diversity in South African infants increased significantly from week 1 to week 15 [median 2.33 (IQR: 1.96–2.62) versus 2.54 (IQR: 2.26–2.77), $P = 0.036$], while α-diversity in Nigerian infants significantly decreased [median 1.13 (IQR: 0.77–1.68) versus 0.87 (IQR: 0.51–1.166), $P < 0.0001$] (Fig. 2A), further exacerbating the differences in α-diversity between sites. There was distinct microbial community composition among Nigerian samples by age, which was less evident for South African infants (Fig. 2B). In agreement, the dominant bacterial changed only marginally from week 1 to week 15 among South African infants, while Nigerian infants experienced a shift from a Firmicutes-dominated microbiota (cluster 3) to one dominated by *Bifidobacterium infantis* and *Streptococcus salivarius* (cluster 2) at 15 weeks of age (Fig. 2C; Fig. 3). Given the differences in delivery mode and proportion of exclusively breastfed infants between sites, we also performed the analysis restricting to EBF ($n = 212$) or vaginally delivered ($n = 249$) infants. The significant differences observed between countries in α- and β-diversity remained over the 15 weeks when the comparison was strictly among EBF infants or vaginally delivered infants (Fig. S2 and S3).

## HIV exposure has a subtle effect on the gut microbiota regardless of the geographical location

There were no significant differences in α-diversity (Fig. S4A), β-diversity (Fig. S4B), or PAM cluster transition (Fig. S4C and D) by HIV exposure status in either country. Differential abundance testing using ANCOM-BC was performed adjusting for feeding mode at the week 15 time point (34). Several bacterial taxa were significantly associated with HIV exposure status in South Africa (Table 2). Several *Enterococcus* species were significantly more abundant in iHEU than iHUU at week 1 (*E. faecium*; LFC: 0.57) and week 15 [*E. faecalis*, *Enterococcus gilvus,* and *Enterococcus raffinosus*; LFC: 0.61, 1.02, and 0.76, respectively). Moreover, *Collinsella aerofaciens* (LFC: 0.72 at week 1 and 1.18 at week 15) and *Klebsiella quasipneumoniae* (LFC: 0.84 at both week 1 and week 15), which are known to be pathobionts (37, 38), were consistently more abundant in iHEU during the first 15 weeks of life. In contrast, no bacterial taxa were differentially abundant by HIV exposure in the Nigerian cohort. To disentangle the effects of co-trimoxazole and HIV exposure on infant gut microbiota, we assessed the gut microbiota based on reported co-trimoxazole prophylaxis history. We did not see any effects of co-trimoxazole on α- and β-diversity among South African iHEU at 15 weeks of age (Fig. S5). We further explored whether co-trimoxazole partially contributed to the differentially enriched bacterial taxa that were identified in South African iHEU. When adjusting the ANCOM-BC for reported co-trimoxazole prophylaxis, several bacterial taxa were no longer enriched in iHEU (Table S2), including *Enterococcus* species (*E. faecalis* and unclassified species), *Veillonella atypica*, and *Staphylococcus* (unclassified species).

## Maternal HIV status and infant gut microbes are associated with infant TT vaccine response

Among the 278 infants, plasma IgG anti-tetanus antibody data were available from 77 South African (59 iHEU and 18 iHUU) and 192 Nigerian infants (138 iHEU and 54 iHUU). In Nigeria, it is recommended that pregnant women receive TT booster vaccinations, whereas this is not policy in the Western Cape, South Africa (39). Therefore, not surprisingly, infant anti-tetanus IgG concentrations in the first week of life, representing

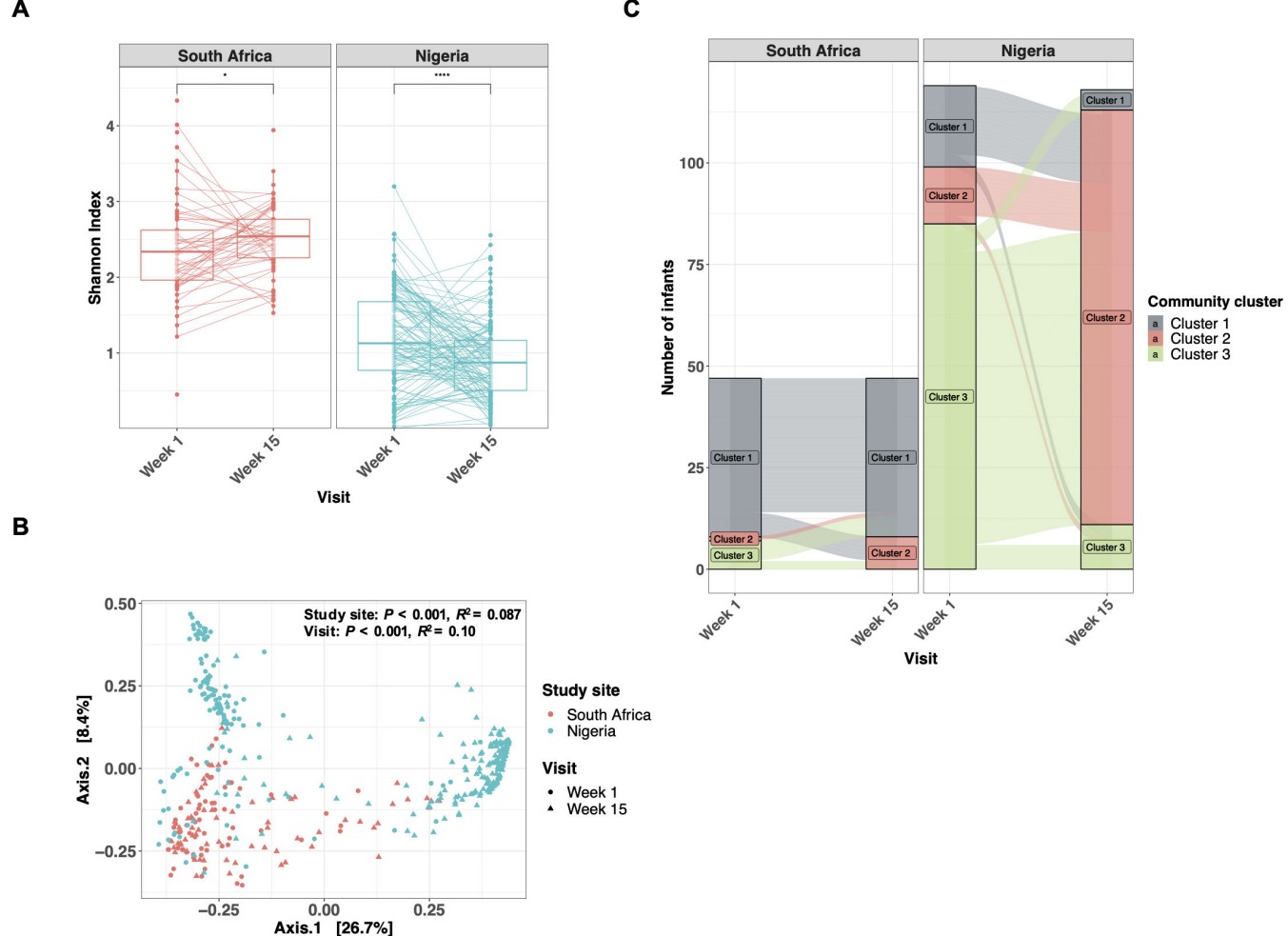

**FIG 2** Longitudinal transition of gut microbiota is distinct among infants in South Africa and Nigeria. (A) The transition of α-diversity (Shannon index) of infant gut microbiota over the first 15 weeks of age in South Africa ($n = 82$) and Nigeria ($n = 196$). (B) PCoA and PERMANOVA (Bray-Curtis dissimilarity) of gut microbiota at 1 week and 15 weeks of age, colored by study site and shaped by visit (South African, $n = 82$; Nigerian, $n = 196$). (C) Alluvial plot showing the transition of cluster groups from week 1 to week 15 at each study site (South African, $n = 82$; Nigerian, $n = 196$). Samples were grouped according to PAM clustering ($k = 3$), indicated by color. *$P < 0.05$; ****$P < 0.0001$.

maternally transferred antibodies, were significantly lower among South African infants than Nigerian infants [median 1.0 (IQR: 0.55–2.2) versus 1.5 (IQR: 0.9–4.1) IU/mL, adj $P$ = 0.002; Fig. S6A]. In contrast, titers did not differ between South African and Nigerian infants at 15 weeks of age [median 1.9 (IQR: 0.65–2.5) versus 1.6 (IQR: 1.0–3.9) IU/mL, adj $P$ = 0.280]. We investigated the correlation of TT vaccine response between mother and infant pairs living in Nigeria ($n = 191$). Anti-tetanus titers were strongly correlated at week 1. However, iHEU mother-infant anti-tetanus titers showed a lower Pearson's correlation coefficient compared to iHUU [$R$: 0.72 ($P < 0.001$) versus 0.95 ($P < 0.001$)] (Fig. 4A). The correlation between maternal and infant anti-tetanus IgG levels was no longer evident by 15 weeks of age in either iHEU or iHUU (Fig. S6B). We did not see any difference in anti-tetanus IgG titers among Nigerian mothers by their HIV status (Fig. S6C). However, iHEU had significantly lower anti-tetanus IgG concentrations than iHUU at 15 weeks of life ($P = 0.016$), and this remained significant after adjusting for multiple comparisons (adj $P = 0.031$; Fig. 4B). The difference between iHEU and iHUU at week 15 was no longer statistically significant when infants were compared separately by study site (adj $P$ = 0.290 in South Africa and adj $P = 0.180$ in Nigeria; Fig. S6D).

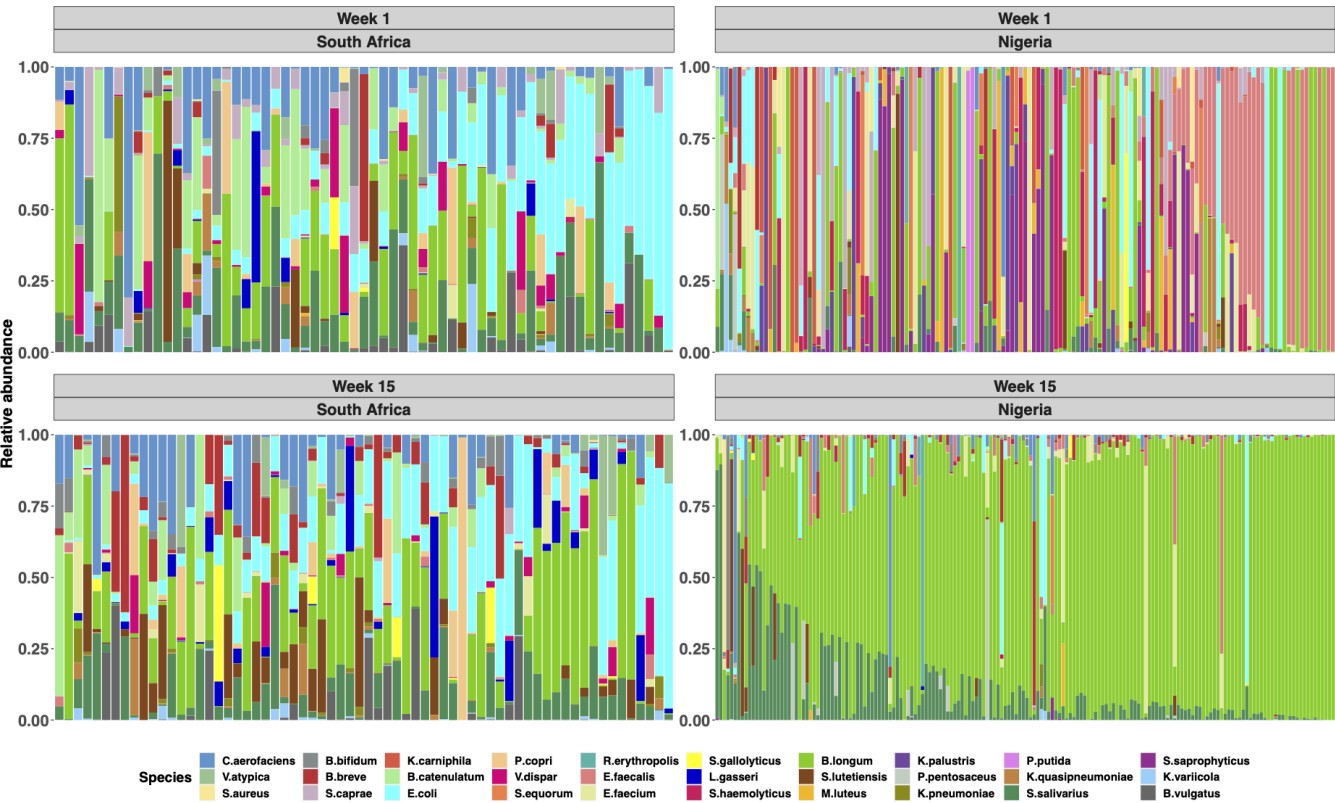

**FIG 3** Infants' gut microbial succession over the first 15 weeks differs substantially between the study sites. Relative abundance plot of most abundant 30 taxa of South African (*n* = 82) and Nigerian (*n* = 196) infants at the 1 week and 15 weeks of age. Each column represents individual participants. *B. longum* subspecies *infantis* and *B. longum* subspecies *longum* are indicated as the same color (green).

Since gut microbiome is thought to modulate the development of the immune system (15), we investigated the relationship between infant gut microbiota and TT vaccine response at week 15, a week after infants complete their primary TT series. We did not see consistent correlations between 15-week anti-tetanus IgG titers and Shannon diversity at 1 or 15 weeks (Fig. 5A). To further explore factors associated with infant TT vaccine response at week 15, we conducted a LASSO regression. Rank-transformed top 50 ASVs at either week 1 or week 15, HIV exposure status, and anti-tetanus IgG titers at week 1 were included as explanatory variables to investigate the predictor, TT vaccine response at 15 weeks of age. In South Africa, infant HIV exposure status showed a strong negative association with 15-week TT vaccine response (β-coefficient = −0.44), and the rank-transformed taxon abundance at week 1 of some bacterial species, including *Streptococcus salivarius* (β-coefficient = 0.038), *Bacteroides dorei* (β-coefficient = 0.016), *Collinsella aerofaciens* (β-coefficient = 0.015), and *Sutterella wadsworthensis* (β-coefficient = −0.011) were independently associated with vaccine response, albeit with weaker β-coefficients than HIV-exposure (Fig. 5B; Table S3A). In contrast, no variables were selected as predictors of the TT vaccine response in the Nigerian cohort. Previously, it has been shown that passively transferred maternal antibody interferes with infant TT vaccination response (40). Since Nigerian infants showed significantly higher maternal antibodies than South African infants at week 1 (Fig. S6A), we speculated that these maternal anti-tetanus antibodies may have masked any associations underlying the infant TT vaccine response at week 15 of life. For this reason, we re-assessed the LASSO regression without including week 1 anti-tetanus IgG data in the explanatory variables (Fig. S7; Table S3B). Although there was no change in the result for the South African infants (Fig. S7A), HIV exposure and several bacteria present at 15 weeks of age, including *S. salivarius*, were independently associated with the TT vaccine response in Nigerian

**TABLE 2** ANCOM-BC analysis of iHEU and iHUU living in South Africa[b]

| Taxonomy (genus, species) | Taxon ID | LFC[a] |
|---|---|---|
| **At 1 week of age** | | |
| *Klebsiella variicola* | ASV46 | 1.22 |
| *Sutterella* (unclassified) | ASV150 | 1.02 |
| *Holdemanella* (unclassified) | ASV53 | 1.00 |
| *Parabacteroides merdae* | ASV101 | 0.98 |
| *Catenibacterium* (unclassified) | ASV218 | 0.96 |
| *Blautia obeum* | ASV59 | 0.93 |
| *Senegalimassilia* (unclassified) | ASV145 | 0.87 |
| *Bifidobacterium breve* | ASV10 | 0.84 |
| *Klebsiella quasipneumoniae* | ASV36 | 0.84 |
| *Libanicoccus* (unclassified) | ASV153 | 0.81 |
| *Blautia* (unclassified) | ASV225 | 0.80 |
| *Ruminococcus torques* group (unclassified) | ASV75 | 0.72 |
| *Collinsella aerofaciens* | ASV25 | 0.72 |
| *Subdoligranulum* (unclassified) | ASV251 | 0.70 |
| *Bacteroides vulgatus* | ASV83 | 0.70 |
| *Sutterella* (unclassified) | ASV496 | 0.67 |
| *Klebsiella pneumoniae* | ASV39 | 0.66 |
| *Megamonas* (unclassified) | ASV169 | 0.65 |
| *Romboutsia ilealis* | ASV93 | 0.64 |
| *Senegalimassilia* (unclassified) | ASV171 | 0.64 |
| *Faecalibacterium* (unclassified) | ASV505 | 0.58 |
| *Fusobacterium mortiferum* | ASV278 | 0.57 |
| *Enterococcus faecium* | ASV7 | 0.57 |
| *Parabacteroides distasonis* | ASV138 | 0.54 |
| *Actinomyces* (unclassified) | ASV668 | −0.54 |
| *Parabacteroides distasonis* | ASV203 | −0.57 |
| **At 15 weeks of age** | | |
| *Streptococcus gallolyticus* | ASV44 | 1.33 |
| *Collinsella aerofaciens* | ASV25 | 1.18 |
| *Clostridium innocuum* group (unclassified) | ASV336 | 1.13 |
| *Enterococcus gilvus* | ASV157 | 1.02 |
| *Klebsiella quasipneumoniae* | ASV42 | 0.84 |
| *Veillonella atypica* | ASV163 | 0.83 |
| *Enterococcus raffinosus* | ASV338 | 0.76 |
| *Enterococcus* (unclassified) | ASV40 | 0.71 |
| *Bifidobacterium adolescentis* | ASV296 | 0.71 |
| *Enterococcus raffinosus* | ASV51 | 0.71 |
| *Lactococcus lactis* | ASV206 | 0.66 |
| *Enterococcus faecalis* | ASV5 | 0.61 |
| *Granulicatella* (unclassified) | ASV681 | 0.59 |
| *Dorea formicigenerans* | ASV229 | 0.59 |
| *Faecalibacterium prausnitzii* | ASV103 | 0.52 |
| *Staphylococcus* (unclassified) | ASV41 | 0.51 |
| *Lactobacillus rhamnosus* | ASV191 | −0.51 |
| *Klebsiella michiganensis* | ASV266 | −0.51 |
| *Lactobacillus gasseri* | ASV161 | −0.53 |
| *Prevotella copri* | ASV405 | −0.55 |
| *Megasphaera elsdenii* | ASV167 | −0.58 |
| *Olsenella* (unclassified) | ASV97 | −0.58 |
| *Prevotella* (unclassified) | ASV176 | −0.83 |
| *Bacteroides caccae* | ASV552 | −0.93 |

(*Continued on next page*)

**TABLE 2** ANCOM-BC analysis of iHEU and iHUU living in South Africa[b] (*Continued*)

| Taxonomy (genus, species) | Taxon ID | LFC[a] |
| --- | --- | --- |
| *Olsenella* (unclassified) | ASV120 | −1.89 |
| *Ruminococcus torques* group (unclassified) | ASV75 | −2.43 |

[a]Abundance in iHEU in relation to iHUU. ANCOM-BC, Analysis of Compositions of Microbiomes with Bias Correction; LFC, log$_e$ fold change; ASV, amplicon sequence variant; iHEU, infants who are HIV-exposed uninfected; iHUU, infants who are HIV-unexposed uninfected.

[b]Differentially abundant ASVs (adj $P$ < 0.05) among iHEU relative to iHUU at 1 week or 15 weeks of age in South Africa ($n$ = 82). Data at week 15 were adjusted by mode of feeding. Positive LFC values indicate higher abundance among iHEU, whereas negative LFC values indicate higher abundance among iHUU. No differentially abundant bacterial taxa were identified among Nigerian infants.

infants (Fig. S7B). However, the β-coefficients for all selected predictors were small, including HIV exposure status.

## DISCUSSION

This is one of the largest studies that longitudinally compared the gut microbiota between iHEU and iHUU in two settings and investigated the association with their vaccine responses. Our findings suggest that the country of origin was the most influential factor in the infants' gut microbiota at week 1, which also strongly affected its succession over the first 15 weeks of life. Both feeding and delivery modes have been shown to influence infant gut microbiota (41). Since these demographic characteristics significantly differed between our South African and Nigerian cohorts, we explored their potential effects on the infant gut microbiota. A comparison restricted to infants still EBF at 15 weeks showed that α- and β-diversity at that time point remained significantly different between the countries. Similarly, the difference was independent of mode of delivery. Analysis of stool samples collected shortly after birth suggested that the difference in gut microbiota profile was already prominent before feeding was established. Collectively, these data indicate that geographical location strongly

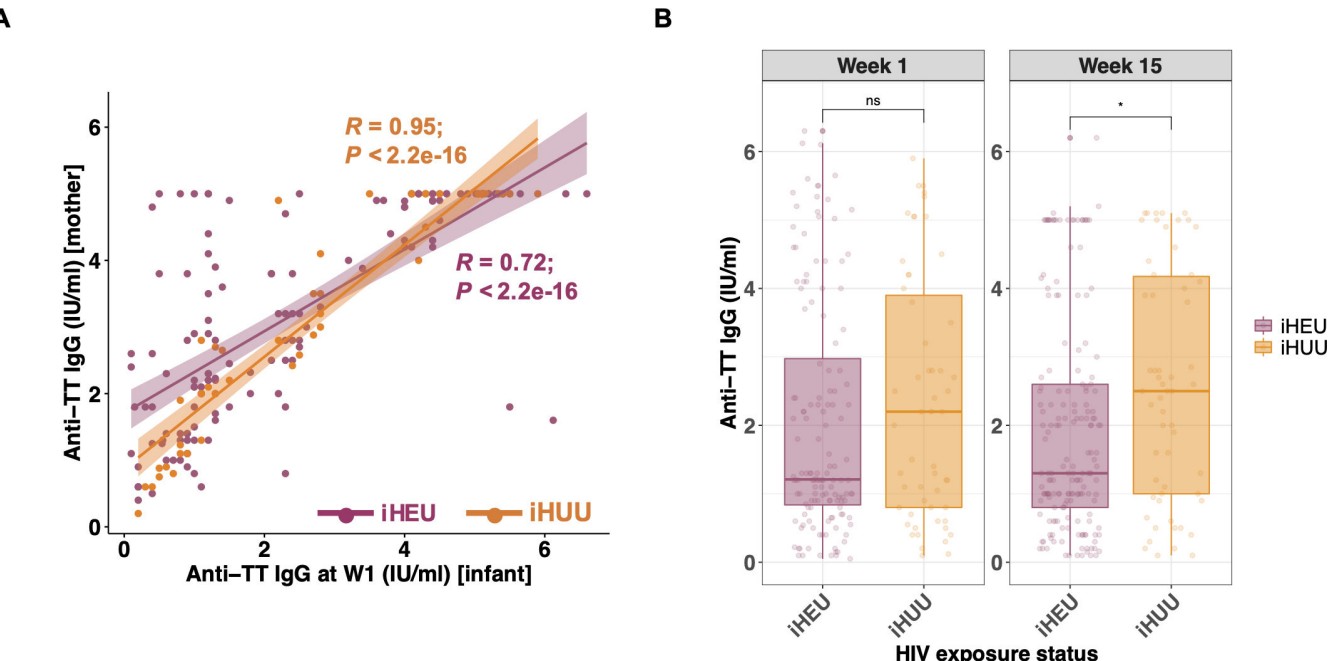

**FIG 4** Passive maternal antibody and HIV exposure are both associated with infant TT vaccine response. (A) Scatter plot and Spearman's rank correlation coefficients ($R$) of anti-tetanus IgG titers (IU/mL) between Nigerian mothers (y-axis; $n$ = 191) and their infants at week 1 (x-axis; $n$ = 191). Dots and lines of best fit are colored by HIV exposure status. (B) Comparison of anti-tetanus IgG titers between iHEU ($n$ = 197) and iHUU ($n$ = 72) at week 1 and week 15. *P*-values comparing anti-tetanus IgG titers were adjusted for multiple comparisons using the Benjamini-Hochberg method. W1, 1 week of age; ns, not significant. *$P$ < 0.05.

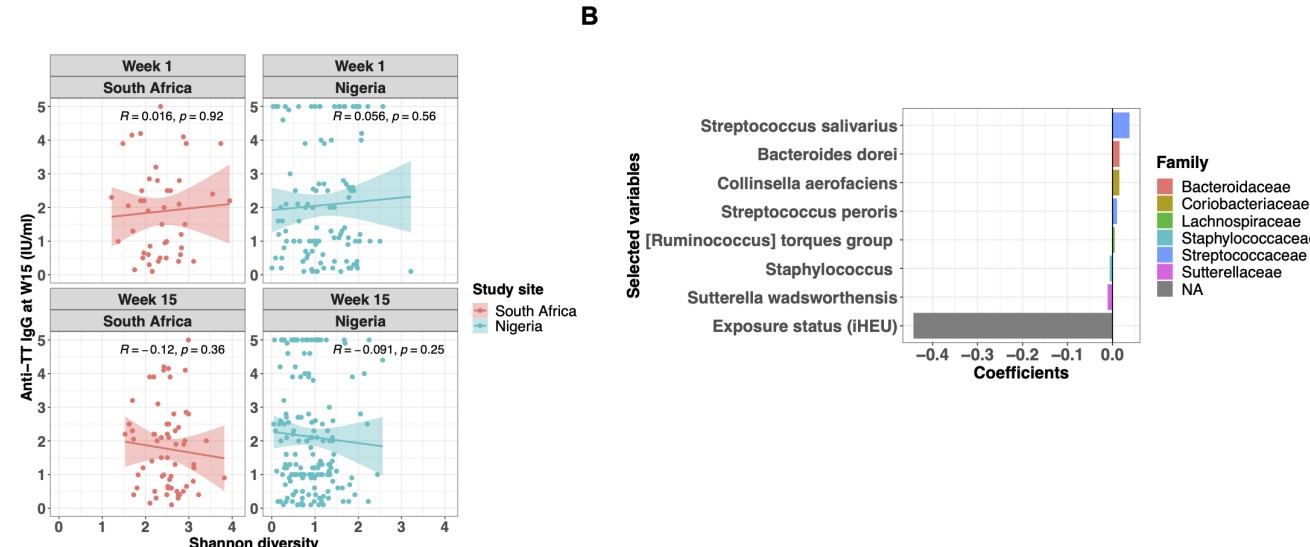

**FIG 5** HIV exposure status and gut microbiota are independently associated with TT vaccine response. (A) Correlation analysis of infants' anti-tetanus IgG titers (IU/mL) measured at 15 weeks of age and α-diversity (Shannon index) at each study site and visit (South African, n = 65; Nigerian, n = 170). Spearman's rank correlation coefficients (R) are indicated on each panel. (B) Rank-transformed top 50 ASVs (at either week 1 or week 15), HIV exposure status, and anti-tetanus IgG titer data at week 1 were used as explanatory variables for the LASSO regression to assess the association with TT vaccine response at 15 weeks of age. Each model was constructed separately based on geographical location and time point. The optimal coefficient tuning parameter (lambda.min) was chosen using 10-fold cross-validation. Selected variables and their glmnet coefficients were plotted. Color of the bars represents taxonomy at the family level. Week 1 ASVs and HIV exposure status were associated with week 15 TT vaccine response among South African infants. No variables were selected for the Nigerian cohort. W15, 15 weeks of age.

influences the initial seeding of gut microbes, and this affects the trajectory of microbiota regardless of feeding practices. Notably, the term "geography" includes not only the physical location (rural versus urban) but also extends to socioeconomics, genetics, diet, climate, and ethnicity, among others.

Microbiota among Nigerian infants transitioned drastically over the first 15 weeks of life, such that at 15 weeks, *B. infantis* was the dominant taxon, with some *S. salivarius*. Both are commonly found in breast milk and gut microbiota among breastfed infants (42). *Bifidobacteria* benefit human health and are often used as probiotics (43). Moreover, *Streptococcus salivarius*, classified as a lactic acid bacterium, has been shown to have probiotic properties (44). However, the drastic changes in microbiota over the 15 weeks only occurred mostly among the Nigerian infants and far less in the South African infants. Plausible explanations for this may be the difference in profiles of maternal gut and breastmilk microbiota and human milk oligosaccharides influenced by genetics, ethnicity, diet, and body mass index (BMI) (45). For instance, a higher maternal BMI is associated with reduced *Bifidobacterium* in breastmilk (46), which might suggest that South African mothers, who had higher mean weight, had less *Bifidobacterium* in their breastmilk than Nigerian mothers, leading to less *Bifidobacterium* in their infants' gut. An additional explanation is that Nigerian mothers in this setting have a diet rich in fermented foods, whereas South African mothers may have a more Westernized diet (47, 48). South African infants also had higher relative abundance of *B. longum* but lower relative abundance of *B. infantis* at baseline.

Nigerian infants had significantly higher anti-tetanus titers at week 1, likely due to maternal immunization and consequent high passive maternal antibody transfer compared to South African infants (49). In contrast, in the Western Cape region where our South African cohort was recruited, there was no routine TT booster vaccination for pregnant women due to the prolonged absence of neonatal tetanus cases in the province (39). Notably, anti-tetanus IgG levels post-vaccination among Nigerian infants remained similar to those at week 1 and were comparable with South African infants.

This inferior induction of anti-tetanus IgG titers observed in Nigerian infants may be explained by the inhibition of TT vaccine response by passively transferred high maternal antibodies, as previously described (40).

We identified several bacterial taxa that exhibited differential abundance in iHEU compared to iHUU, including several pathobionts. Whether these identified bacterial taxa contribute to increased risk of infectious morbidity in iHEU is unknown. Moreover, several bacterial enrichments in iHEU may, in part, be attributed to co-trimoxazole prophylaxis recommended for this population. For example, *E. faecalis* frequently carries co-trimoxazole resistance (50). Supporting this, our differential abundance analysis showed that the enrichment of *E. faecalis* was no longer evident in 15-week-old iHEU after adjusting for reported co-trimoxazole prophylaxis use. Of note, there was a significant difference in reported co-trimoxazole use among iHEU between the countries in our study. Since the record of co-trimoxazole treatment was sorely relied on mothers' recall at each follow-up visit, it is possible that the accuracy of reported antibiotics records among iHEU may be underestimated.

LASSO regression models also suggested that *in utero* HIV exposure and relative abundance of several bacterial taxa at week 1 were independently associated with later TT vaccine response in South Africa but not Nigeria. The higher passive antibody levels observed in Nigerian infants may have mitigated the effects of HIV exposure and microbiota on the infant vaccine response. In fact, excluding the week 1 titer data from the regression model indicated that the HIV exposure and several microbes found at 15 weeks of age were independently associated with the infant TT vaccine response among Nigerian infants. In line with our assumption, removing week 1 titer data from the regression model did not change the LASSO regression result in South African infants, who showed much lower passive maternal antibody transfer. Interestingly, *S. salivarius* relative abundance was predictive of improved anti-tetanus IgG titers in both cohorts. Since the microbiota at week 1 among South African infants was associated with TT vaccine response at week 15, this suggests that in some settings, vaccine responses could potentially be modified using an early-life microbiome intervention where maternal vaccination is not possible.

There are several limitations in our study. Firstly, we did not have comprehensive records of maternal lifestyle and dietary information, which are known to have an impact on gut microbiota composition. In addition, co-trimoxazole adherence among iHEU was not extensively captured. All iHEU received nevirapine post-exposure prophylaxis; therefore, the effects of HIV exposure versus antiretroviral exposure cannot be disentangled. Lastly, additional data, both maternal (such as vaginal and breastmilk microbiota) and infant (such as gut metabolomics, metatranscriptomics, and metagenomics), could have provided more insights into our study.

## Conclusions

This study showed that the transition of infant gut microbiota was strongly dependent on geographical location and age, while effect of *in utero* HIV exposure was modest. However, maternal HIV status was negatively associated with the passive maternal anti-tetanus antibody transfer, and the negative effect of HIV exposure on TT vaccine response persisted over the first 15 weeks of life among iHEU. In addition, there were independent associations of specific gut microbes and HIV exposure with infant humoral response to TT vaccine at 15 weeks of age.

### ACKNOWLEDGMENTS

We would like to thank the mothers and infants who participated in this study, the clinic staff at the Plateau State Specialist Hospital, Jos, Nigeria, and the Khayelitsha Site B Clinic in Cape Town, South Africa, and the InFANT study lab technologists.

This work was funded by the Canadian Institutes of Health Research (CIHR) (01044–000 awarded to C.M.G.) and the U.S. National Institutes of Health (NIH), Eunice Kennedy

Shriver National Institute of Child Health and Human Development (NICHD), and the National Human Genome Research Institute (NHGRI) (U01HD094658 to A.A.). S.C.I. was funded by the Yoshida Scholarship Foundation.

C.M.G., A.A., J.M.B., and H.B.J. contributed to the conception and design of the study. S.O. was responsible for the acquisition of anti-tetanus IgG data. S.C.I. was responsible for acquiring and analyzing microbiome data and making graphs, tables, and additional files. S.C.I., H.B.J., A.H., and S.P.H. were involved in data interpretation. S.C.I., H.B.J., and A.H. wrote the manuscript. All authors participated in drafting and revising the manuscript and approved the final version of the manuscript.

## AUTHOR AFFILIATIONS

[1]Division of Immunology, Department of Pathology, University of Cape Town, Cape Town, South Africa

[2]Institute of Infectious Disease and Molecular Medicine, University of Cape Town, Cape Town, South Africa

[3]Institute of Human Virology-Nigeria, Abuja, Nigeria

[4]Division of Molecular Biology and Human Genetics, Biomedical Research Institute, Stellenbosch University, Cape Town, South Africa

[5]Department of Statistics, Stanford University, Stanford, California, USA

[6]Division of Chemical and Systems Biology, University of Cape Town, Cape Town, South Africa

[7]Institute of Human Virology, University of Maryland, School of Medicine, Baltimore, Maryland, USA

[8]Seattle Children's Research Institute, Center for Global Infectious Disease Research, Seattle, Washington, USA

## AUTHOR ORCIDs

Saori C. Iwase  http://orcid.org/0009-0009-6481-0305
Sophia Osawe  http://orcid.org/0000-0001-7998-9680
Anna-Ursula Happel  http://orcid.org/0000-0003-3658-8638
Heather B. Jaspan  http://orcid.org/0000-0002-0745-6073

## FUNDING

| Funder | Grant(s) | Author(s) |
|---|---|---|
| HHS \| NIH \| Eunice Kennedy Shriver National Institute of Child Health and Human Development (NICHD) | U01HD094658 | Alash'le Abimiku |
| Gouvernement du Canada \| Canadian Institutes of Health Research (IRSC) | OCB034114 | Heather B. Jaspan |
| Yoshida Scholarship Foundation (YSF) | | Saori C. Iwase |
| HHS \| NIH \| National Institute of Allergy and Infectious Diseases (NIAID) | R01AI120714 | Heather B. Jaspan |

## AUTHOR CONTRIBUTIONS

Saori C. Iwase, Data curation, Formal analysis, Investigation, Visualization, Writing – original draft | Sophia Osawe, Data curation, Formal analysis, Supervision, Writing – review and editing | Anna-Ursula Happel, Data curation, Formal analysis, Investigation, Supervision, Writing – review and editing | Clive M. Gray, Conceptualization, Writing – review and editing | Susan P. Holmes, Methodology, Supervision, Writing – review and editing | Jonathan M. Blackburn, Conceptualization, Writing – review and editing | Alash'le Abimiku, Conceptualization, Funding acquisition, Supervision, Writing – review and editing.

## DATA AVAILABILITY

Sequencing reads from the 16S rRNA gene profiling are available at the Sequence Read Archive repository hosted by NCBI (accession number PRJNA976299). Scripts and related data sets can be found at https://github.com/saiwase/BEAMING_GUT_MICROBIOME.

## ETHICS APPROVAL

The study was conducted under the Declaration of Helsinki. The protocol was approved by the Ethics Committee of the University of Cape Town (HREC 285/2012), the Institutional Review Board of Seattle Children's Hospital (Study # 15690), and the Human Research Committee of the Plateau State Hospital Jos (PSSS/ADM/ETHC/2015/004). Written informed consent was obtained from all mothers in their preferred language prior to enrolment.

## ADDITIONAL FILES

The following material is available online.

### Supplemental Material

**Fig. S1 (Spectrum03190-23-s0001.pdf).** α-diversity of meconium samples differs significantly by study site.
**Fig. S2 (Spectrum03190-23-s0002.pdf).** α- and β-diversity significantly differ between the countries in exclusively breastfed infants.
**Fig. S3 (Spectrum03190-23-s0003.pdf).** α- and β-diversity significantly differ between the countries in vaginally delivered infants.
**Fig. S4 (Spectrum03190-23-s0004.pdf).** HIV exposure has a subtle effect on gut microbiota across two African countries.
**Fig. S5 (Spectrum03190-23-s0005.pdf).** The effect of co-trimoxazole on α- and β-diversity of gut microbiota is marginal.
**Fig. S6 (Spectrum03190-23-s0006.pdf).** Association of infant anti-tetanus titer with age, HIV exposure status and mother's anti-tetanus titer.
**Fig. S7 (Spectrum03190-23-s0007.pdf).** Maternal antibodies may mask the effect of HIV exposure and microbiota on infant vaccine response.
**SupplementalMaterials (Spectrum03190-23-s0008.docx).** Supplemental tables and figure legends.

### Open Peer Review

**PEER REVIEW HISTORY (review-history.pdf).** An accounting of the reviewer comments and feedback.

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
