## [Reviewer comments · Microbiology Spectrum]

Microbiology Spectrum

Longitudinal gut microbiota composition of South African and Nigerian infants in relation to tetanus vaccine responses

Saori Iwase, Sophia Osawe, Anna Happel, Clive Gray, Susan Holmes, Jonathan Blackburn, Alash'le Abimiku, and Heather Jaspan

Corresponding Author(s): Heather Jaspan, University of Washington

Review Timeline:

Submission Date:	August 29, 2023
Editorial Decision:	October 5, 2023
Revision Received:	December 4, 2023
Accepted:	December 20, 2023

Editor: Laxmi Yeruva

Reviewer(s): The reviewers have opted to remain anonymous.

Transaction Report:

DOI: <https://doi.org/10.1128/spectrum.03190-23>

October 5, 2023

Dr. Heather B Jaspan
University of Washington
Department of Pediatrics and Global Health
Seattle, WA

Re: Spectrum03190-23 (Longitudinal gut microbiota composition of South African and Nigerian infants in relation to tetanus vaccine responses)

Dear Dr. Heather B Jaspan:

Link Not Available

Sincerely,

Laxmi Yeruva

Journals Department
Reviewer comments:

Reviewer #2 (Comments for the Author):

The study is interesting and structured well. However, there is a lack of clarity on the number of samples used in the study. For example, it was stated that 47 South African infants' gut microbiota information was available at both time points (1st and 15th week), but there needed to be more information on how many were breastfed or mixed-fed. Further, the type of feeding mode significantly alters the infant gut microbiota; more information on the transition time from breast milk to mixed feeding in South African infants would be helpful for the reader.

Staff Comments:

Preparing Revision Guidelines

Please return the manuscript within 60 days; if you cannot complete the modification within this time period, please contact me. If you do not wish to modify the manuscript and prefer to submit it to another journal, please notify me of your decision immediately so that the manuscript may be formally withdrawn from consideration by Microbiology Spectrum.

In the present manuscript, “Longitudinal gut microbiota composition of South African and Nigerian infants in relation to tetanus vaccine responses”, the authors studied the gut microbiome of infants from two African countries who were either exposed to HIV but not infected or those non-exposed and not infected to HIV and evaluated if the HIV exposure impacts the infants’ gut microbiome and response to tetanus toxoid vaccine responses. The authors showed that geographical location and age had a more robust impact on the gut microbiota composition than exposure to HIV. Using the Nigerian cohort, they also showed that the passive transfer of maternal anti-Tetanus Toxoid antibody might mitigate the association between HIV exposure or certain gut microbiota taxa with TT vaccine response.

Major comment:

It is unclear if the authors considered anti-retroviral post-exposure prophylaxis at birth and cotrimoxazole use in iHEU infants starting 6 weeks of age (lines 93-94) when gut microbiota was analyzed longitudinally. Since cotrimoxazole is a broad-spectrum antibiotic, it is important to consider how the use of this antibiotic impacts the succession of the gut microbiota at a later age (15 weeks in this study). In the South African cohort, several *Enterococcus* species were significantly more abundant in iHEU than iHUU at week 15 (lines 226-228). Is it the effect of HIV exposure or the use of Cotrimoxazole in infants (since several species of *Enterococcus* species are considered antibiotic-resistant)?

Minor comments:

- 1) Line 200 – (1) “Actinobacteriota including several *Bifidobacterium*”:
Comment: Do the authors mean Actinobacteria (old name) or Actinomycetota (new name)?

- 2) Line 227 – “*Enterococcus* species (*E. faecalis*, *E. faecium*, *E. gilvus*, and *E. raffinosus*) were significantly more abundant in iHEU than iHUU at week 15...”

Comment: *E. faecium* is more in iHEU than iHUU at week 1, not week 15 (Supplementary Table 2).

- 3) Lines 288 – 291: the references provided do not seem relevant here.

1. Line 29: Please include the number of samples under the iHEU and iHUU categories in the abstract.
2. Line 107: Please provide the details of the manufacturer of the booster TT vaccine.
3. Line 131: Please include the procedure of extraction of plasma from the collected blood samples.
4. Line 133: Please indicate the sample size of the subset along with their HIV status.
5. Line 105, 136: The coefficient of determination (r^2) value of the standard curve needs to be included. Also, please indicate the intra and inter-assay coefficient of variance value of the kit.
6. Line 188-189: According to the results, 278 mother and infant pairs were included, and the samples were collected at 1 and 15- week intervals, which makes the total number of samples to be 556. However, it is mentioned that a total of 524 samples were used for the analyses. Please provide an explanation for this difference in sample numbers.
7. Line 221: Please indicate the sample numbers for EBF and vaginally delivered infants.
8. Line 229-231: A brief discussion on the association of microbial genera and HIV status might be interesting.
9. The bibliography sections need to follow one uniform format accepted by the journal guidelines. For example, in line 539, the spacing between L and V in Blanton L V., and the punctuation after V need to be removed.
10. Please include limitations of the study; for example, the lifestyle (sedentary or moderate or heavy working) and diet (meat-eating or vegan or vegetarian, etc.) of the pregnant mothers can influence the gut microbiome, subsequently altering the immune responses.

REVIEWER 1

Major comment: It is unclear if the authors considered anti-retroviral post-exposure prophylaxis at birth and cotrimoxazole use in iHEU infants starting 6 weeks of age (lines 93-94) when gut microbiota was analyzed longitudinally. Since cotrimoxazole is a broad-spectrum antibiotic, it is important to consider how the use of this antibiotic impacts the succession of the gut microbiota at a later age (15 weeks in this study).

Response: Thank you for raising this critical point. We agree that the effect of antibiotics on gut microbiota is an important factor to consider. It is possible that postexposure prophylaxis may also alter gut microbiota, but all iHEU received nevirapine, therefore the effects of HIV-exposure and ART exposure cannot be disentangled. We included information about antibiotics history in revised Table 1 and Supplementary Table S1. We also had additional sentences in Results (lines 194-197) and Discussion section (lines 350-353) in the revised manuscript as follows:

[Results] *“History of antibiotic use was higher among iHEU due to co-trimoxazole prophylaxis (86.6% iHEU versus 6.6% iHUU, $P < 0.001$). Significantly fewer South African iHEU reported co-trimoxazole prophylaxis than Nigerian iHEU (55.7% versus 100%, $P < 0.001$).”*

[Discussion] *“Of note, there was a significant difference in reported co-trimoxazole use among iHEU between the countries in our study. Since the record of co-trimoxazole treatment was solely relied on mothers’ recall at each follow-up visit, it is possible that the accuracy of reported antibiotics records among iHEU may be underestimated.”*

Moreover, we included an additional analysis of the effect of reported co-trimoxazole use on alpha- and beta-diversity of gut microbiota among 15-week-old South African iHEU. These results are shown in Supplementary Figure S5 and mentioned in the revised manuscript as follows (lines 251-253):

[Results] *“We did not see any effects of co-trimoxazole treatment on α - and β -diversity among South African iHEU at 15 weeks of age (Supplementary Figure S5).”*

For the effects of PEP, we have made the following changes (lines 96 and 370-372):

[Methods] We have changed line 96 to read: *“all iHEU received nevirapine post-exposure prophylaxis (PEP)”*.

[Discussion] We have added to the limitations *“All iHEU received nevirapine post-exposure prophylaxis, therefore the effects of HIV exposure versus antiretroviral exposure cannot be disentangled.”*

In the South African cohort, several Enterococcus species were significantly more abundant in iHEU than iHUU at week 15 (lines 226-228). Is it the effect of HIV exposure or the use of Cotrimoxazole in infants (since several species of Enterococcus species are considered antibiotic-resistant)?

Response: We appreciate this comment. To reflect this feedback, we re-assessed ANCOM-BC, adjusting for co-trimoxazole use. The result was included in Supplementary Table S2 in the revised

manuscript and described in both Results (lines 253-257) and Discussion sections (lines 348-352) as follows:

[Results] “We further explored whether co-trimoxazole partially contributed to the differentially enriched bacterial taxa that were identified in South African iHEU. When adjusting the ANCOM-BC for reported co-trimoxazole prophylaxis, several bacterial taxa were no longer enriched in iHEU (Supplementary Table S2), including *Enterococcus* species (*E. faecalis* and unclassified species), *Veillonella atypica*, and *Staphylococcus* (unclassified species).”

[Discussion] “Moreover, several bacterial enrichments in iHEU may, in part, be attributed to co-trimoxazole prophylaxis recommended for this population. For example, *E. faecalis* is frequently co-trimoxazole resistant (50). Supporting this, our differential abundance analysis showed that the enrichment of *E. faecalis* was no longer evident in 15-week-old iHEU after adjusting for reported co-trimoxazole prophylaxis use.”

Minor comments:

1) Line 200 – (1) “Actinobacteriota including several *Bifidobacterium*”: Comment: Do the authors mean Actinobacteria (old name) or Actinomycetota (new name)?

Response: Thank you for pointing this out. We have changed the word to “Actinomycetota” instead of Actinobacteriota (lines 215 and 220).

2) Line 227 – “*Enterococcus* species (*E. faecalis*, *E. faecium*, *E. gilvus*, and *E. raffinosus*) were significantly more abundant in iHEU than iHUU at week 15...” Comment: *E. faecium* is more in iHEU than iHUU at week 1, not week 15 (Supplementary Table 2).

Response: Thank you for pointing this out. We have amended this as follows (lines 244-246).

[Results] “Several *Enterococcus* species were significantly more abundant in iHEU than iHUU at week 1 (*E. faecium*; Log_e fold change (LFC): 0.57) and week 15 (*E. faecalis*, *E. gilvus*, and *E. raffinosus*; LFC: 0.61, 1.02, and 0.76, respectively).”

3) Lines 288 – 291: the references provided do not seem relevant here.

Response: We removed these references from the revised manuscript as suggested by the reviewer (lines 316-318).

REVIEWER 2

The study is interesting and structured well. However, there is a lack of clarity on the number of samples used in the study. For example, it was stated that 47 South African infants’ gut microbiota information was available at both time points (1st and 15th week), but there needed to be more information on how many were breastfed or mixed-fed.

Response: We have indicated sample numbers more clearly in the revised manuscript. Please see our response to the comment #7 below.

Further, the type of feeding mode significantly alters the infant gut microbiota; more information on the transition time from breast milk to mixed feeding in South African infants would be helpful for the reader.

Response: We appreciate this suggestion. More detail regarding the “mixed feeding” group was added in the revised manuscript as follows (lines 190-193):

[Results] *“Among the South African mothers who reported “mixed feeding”, 58.8% (n = 19) introduced formula feeding or solid food while continuing breastfeeding, and 41.2% (n = 14) completely switched to formula feeding during the course of the study (median breastfeeding duration: 32 days).”*

1. Line 29: Please include the number of samples under the iHEU and iHUU categories in the abstract.

Response: We amended the sentence to address the comment raised by the reviewer as follows (lines 29-30):

[Abstract, Methods] *“We evaluated the gut microbiota of 82 South African (61 iHEU and 21 iHUU) and 196 Nigerian (141 iHEU and 55 iHUU) infants at < 1 and 15 weeks of life by 16S rRNA gene sequencing.”*

2. Line 107: Please provide the details of the manufacturer of the booster TT vaccine.

Response: We included the detail of the TT booster given to the pregnant women in the revised manuscript as follows (line 110):

[Methods] *“Pregnant mothers were given booster TT vaccination (Serum Institute of India Pvt. Ltd.) in Nigeria.”*

3. Line 131: Please include the procedure of extraction of plasma from the collected blood samples.

Response: As suggested by the reviewer, we added the procedure of blood collection and plasma isolation procedures as follows (lines 134-137):

[Methods] *“Blood samples were obtained from infants at 1 and 15 weeks and from Nigerian mothers at 1 week postpartum. All blood samples were collected in heparinized tubes and transported to the lab within 6 hours for sample processing. Plasma was removed prior to cell isolation using Ficoll density-gradient separation medium (Sigma Aldrich) and stored at -80°C until analysis.”*

4. Line 133: Please indicate the sample size of the subset along with their HIV status.

Response: Thank you for this suggestion. We described this in the revised manuscript (lines 260-261 and 267-268):

[Results] *“Among the 278 infants, plasma IgG anti-tetanus antibody data was available from 77 South African (59 iHEU and 18 iHUU) and 192 Nigerian infants (138 iHEU and 54 iHUU).”*

“We investigated the correlation of TT vaccine response between mother and infant pairs living in Nigeria (n = 191).”

5. Line 135, 136: The coefficient of determination (r²) value of the standard curve needs to be included. Also, please indicate the intra and inter-assay coefficient of variance value of the kit.

Response: We added the suggested content to the revised manuscript as follows (lines 141-144):

[Methods] *“To validate each assay, we considered only calibration curves for each plate with a coefficient of determination (r²) above 0.95. Calibration curves that had an r² below 0.95 were repeated. The manufacturer provided the intra-assay and inter-assay coefficient of variation (CV%) as 6.9 and 10.4, respectively.”*

6. Line 188-189: According to the results, 278 mother and infant pairs were included, and the samples were collected at 1 and 15-week intervals, which makes the total number of samples to be 556. However, it is mentioned that a total of 524 samples were used for the analyses. Please provide an explanation for this difference in sample numbers.

Response: We indeed included a total of 278 mother-infant pairs in this study. However, not all infants had both week 1 and week 15 stool samples available. For this reason, 524 samples were used for 16S rRNA gene sequencing instead of 556 samples. Among the 524 sequenced samples, 442 passed the quality filtering and were used for the downstream analysis. The diagram below illustrates the breakdown of 16S rRNA samples used for the analysis.

7. Line 221: Please indicate the sample numbers for EBF and vaginally delivered infants.

Response: Thank you for your feedback. We indicated the infant number as follows (lines 233-235):

[Results] *“Given the differences in delivery mode and proportion of exclusively breastfed infants between sites, we also performed the analysis restricting to EBF (n = 212) or vaginally delivered (n = 249) infants.”*

8. Line 229-231: A brief discussion on the association of microbial genera and HIV status might be interesting.

Response: Thank you for the suggestion. We added a new paragraph in the Discussion section to address this point as follows (lines 344-350):

[Discussion] *“We identified several bacterial taxa that exhibited differential abundance in iHEU compared to iHUU, including several pathobionts. Whether these identified bacterial taxa contribute to increased risk of infectious morbidity in iHEU is unknown. Moreover, several bacterial enrichments in iHEU may, in part, be attributed to the co-trimoxazole prophylaxis recommended for this population. For example, E. faecalis is frequently resistant to co-trimoxazole (50). Supporting this, our differential abundance analysis showed that the enrichment of E. faecalis was no longer evident in 15-week-old iHEU after adjusting for reported antibiotic use.”*

9. The bibliography sections need to follow one uniform format accepted by the journal guidelines. For example, in line 539, the spacing between L and V in Blanton L V., and the punctuation after V need to be removed.

Response: Thank you for pointing this out. We have addressed this issue, and the references are listed in a consistent manner.

10. Please include limitations of the study; for example, the lifestyle (sedentary or moderate or heavy working) and diet (meat-eating or vegan or vegetarian, etc.) of the pregnant mothers can influence the gut microbiome, subsequently altering the immune responses.

Response: Thank you for your feedback. We have added an additional paragraph to describe the limitations of our study as follows (lines 368-374):

[Discussion] *“There are several limitations in our study. Firstly, we did not have comprehensive records of maternal lifestyle and dietary information, which are known to have an impact on gut microbiota composition. In addition, co-trimoxazole adherence among iHEU was not extensively captured. All iHEU received nevirapine post-exposure prophylaxis, therefore the effects of HIV exposure versus antiretroviral exposure cannot be disentangled. Lastly, additional data, both maternal (such as vaginal and breastmilk microbiota) and infant (such as gut metabolomics, metatranscriptomics, and metagenomics), could have provided more insights into our study.”*

Lastly, we would like to draw attention to other amendments made in the revised manuscript. These include:

- 1) We have changed the “anti-TT IgG titer” to “anti-tetanus IgG titer” in the revised manuscript. Examples of this change can be seen in lines 30, 137, 163-164, etc.
- 2) We amended the Abstract sentences to provide more context for this study within the word count limit. The word count of the Abstract now stands at 243 words instead of 205 words.
- 3) We added median and IQR values for alpha-diversity and anti-tetanus titers to provide more information to the readers (204, 226-228, 265, and 267).
- 4) We indicated the number of meconium samples included in the analysis (line 213).
- 5) We indicated the number of samples in the figure legends, wherever applicable.
- 6) We included the beta-diversity plot in Supplementary Figure S1B as this plot was missing in the original manuscript.

Re: Spectrum03190-23R1 (Longitudinal gut microbiota composition of South African and Nigerian infants in relation to tetanus vaccine responses)

Dear Dr. Heather B Jaspan:

Your manuscript has been accepted, and I am forwarding it to the ASM production staff for publication. Your paper will first be checked to make sure all elements meet the technical requirements. ASM staff will contact you if anything needs to be revised before copyediting and production can begin. Otherwise, you will be notified when your proofs are ready to be viewed.

Sincerely,
Laxmi Yeruva
Editor
Microbiology Spectrum

Reviewer #1 (Comments for the Author):

The manuscript is well-written. Thank you for addressing my comments on the previous version.

Reviewer #2 (Comments for the Author):

No comments

All my comments for the previous version of the manuscript have been addressed.

My only comment is for Figure 1 A legend. The legend says the top 20 taxa, but there are 18 taxa in the figure. I am not sure if that is top 20 or top 18.

Thank you for the opportunity to review this manuscript.